# Cu(Proline)_2_ Complex: A Model of Bio-Copper Structural Ambivalence

**DOI:** 10.3390/molecules27185846

**Published:** 2022-09-09

**Authors:** Victor V. Volkov, Riccardo Chelli, Carole C. Perry

**Affiliations:** 1Interdisciplinary Biomedical Research Centre, School of Science and Technology, Nottingham Trent University, Clifton Lane, Nottingham NG11 8NS, UK; 2Dipartimento di Chimica, Università di Firenze, via della Lastruccia 3, 50019 Sesto Fiorentino, Italy

**Keywords:** structure, copper, infrared, optical activity, density functional theory

## Abstract

Complexes of Cu^2+^(d^9^) with proline may be considered a simple model to address the structural flexibility and electronic properties of copper metalloproteins. To discuss optical electronic spectra and infrared spectral responses, we use quantum chemistry applied to model systems prepared under different geometries and degree of hydration. A comparison of experimental data with calculations indicates that first explicit neighbor water clustering next to the Cu^2+^(d^9^) complex is critical for a correct description of the electronic properties of this system. We deduce that the moderately hydrated *trans* conformer is the main structural form of the complex in water. Further, we suggest that the antisymmetric stretching mode of the carbonyl moieties of the conformer is dominant in the spectrally broadened infrared resonance at 1605 cm^−1^, where inhomogeneity of the transition at the blue side can be ascribed to a continuum of less optimal interactions with the solvent. Extracted structural properties and hydration features provide information on the structural flexibility/plasticity specific to Cu^2+^(d^9^) systems in correlation with the electronic behavior upon photoexcitation. We discuss the role and the nature of the axial ligand in bio-copper structural ambivalence and reactivity.

## 1. Introduction

Abundant in Earth’s crust [1], copper (particularly, Cu^2+^) is known to be widely involved in structural genesis and in energy flow processes in both inorganic [2] and living matter [3]. The d^9^ electronic configuration of Cu^2+^ is the basis for Jahn–Teller distortion and gives rise to the structural flexibility of coordination compounds of Cu^2+^(d^9^), providing, for example, fast ligand exchange reactivity [4], ligand rearrangements in aqueous systems [5,6], and charge transfer in *σ*-complexes [7].

In living matter, copper proteins provide electron transfer, oxygen transport, reduction, and substrate activation [8,9]. Additionally, within the last decade, there is an increasing awareness of the role of Cu^2+^ as a structural chaperone [10,11,12]. Due to its electronic structure, when in proteins, copper demonstrates a wide structural variance in coordination associated with peculiar structural flexibility. For example, copper electron transfer proteins with reduction potentials ranging from 183 to 800 mV include trigonal pyramidal and distorted tetrahedral (in Type 1 proteins), distorted tetragonal (in Type 2), and tetragonal and trigonal planar (Type 3) geometries [13]. Previous studies have also reported a tetragonal structure at the copper site in oxygen transporting hemocyanin; a tetrahedral arrangement in Type 1 copper nitrite reductase; a square planar geometry in Type 2 copper oxidases; square pyramidal coordination in Type 2 oxidases, hydroxylases, and superoxide dismutase; and a tetragonal structure in Type 3 hydroxylases [9,14,15]. It is important to note that, while copper is second in abundance after iron in metalloproteins, there is a conceptual difference in coordination and structural flexibility in both systems. While coordination in Fe^3+^(d^5^)/Fe^2+^(d^6^) systems can be conserved, the Cu^2+^(d^9^)/Cu^1+^(d^10^) redox couple experiences structural tensions that are intimately linked with the bioreactivity of the metal. Computational studies of several copper proteins confirmed large fluctuations in the computed metal−ligand bond lengths and revisited the role of the entatic contribution to protein reactivity [16].

Given the relative structural flexibility specific to Cu^2+^(d^9^)/Cu^1+^(d^10^) bioreactivity, it is important to consider the entatic model of enzyme reactivity [17]. In particular, the hypothesis suggests that a protein environment may have the capacity to alter the ligand coordination of the reactant system at the active site. The alteration would shift the energy of the reactant higher to make it closer to the energy of the transition state, and hence reduce the activation barrier of the reaction. It was proposed that an effective enzyme would have a structure such that its readjustment (along its numerous degrees of freedom) would compensate the preparation of the higher energetic-stressed (entatic) reactant state [18]. In respect to flexibility specific to Cu^2+^(d^9^)/Cu^1+^(d^10^) bioreactivity, the authors considered that since ”copper(I) and copper(II) demand different site symmetries, i.e., tetrahedral and tetragonal—therefore, a suitable intermediate of irregular symmetry would result in an entatic state”. Within the past decades, however, except for a report by Rorabacher’s group [13], the hypothesis has not found clear experimental evidence for the preparation of a higher energy reactant state compensated by “supplementary” structural redistributions at the metal center. An alternative view of enzyme activity involving an ‘ecstatic’ state, where the probability of reactivity is according to the overlap (in the configurational space) of a structurally floppy reactant and product states, has been championed based on the width of electron paramagnetic resonance spectra of frozen solutions of metalloproteins [19]. However, because the spectral inhomogeneity of frozen enzymes may not relate to reactivity in vivo quantitatively, it is clear that careful sorting of the ‘entatic’ and ‘ecstatic’ contributions to reactivity requires further experimental and theoretical effort.

In addition to roles in charge transfer and catalytic processes, copper cations are reported to play a role in molecular chaperones [10,11,12] and protein misfolding [20,21,22]. For the latter, the same Jahn–Teller related phenomena and structural flexibility are proposed to provide a competitively high affinity for copper binding and structural variance as for amyloid formation [21]. Due to the presence of multiple binding sites [20] and complexity in local structural variances, understanding the structural and dynamic properties of such systems is a challenge. 

In this contribution, following an early suggestion to approach the understanding of structure and reactivity in proteins using simple complexes [23], we test experimental and theoretical approaches to explore the structural and dynamic properties of a Cu(L-proline)_2_ complex, the synthesis of which and preliminary optical characterizations were reported previously [24]. Early efforts in synthesis and characterization of Cu^2+^ complexes with amino acids indicated sensitivity of the optical activity of d-d transitions to the nature of the ligands [25,26]. Subsequent studies mainly concerned assessing whether the optical activity of Cu^2+^ complexes with amino acids in the visible (VIS) spectral range could help to distinguish the chirality of the ligands [27,28,29].

Early interpretations of the observed electronic properties mainly relied on the results of X-ray studies in single crystals [30,31]. Further, the electron spin resonance spectroscopy of complexes of Cu^2+^ with amino acids in solution suggested equilibrium mixtures of structures with N and O atoms in *cis* and *trans* arrangements with respect to the central Cu^2+^ ion [32]. Here, it is important to stress that optical spectroscopy techniques are particularly helpful in the characterization of the structure and dynamics in liquids at room temperature. In this respect, it is interesting to note that infrared (IR) absorption of the carbonyl of a Cu(L-proline)_2_ complex in solution revealed a line-shape that is broadened at the higher frequency side [24], which could be due to: (1) a frequency split between symmetric and antisymmetric carbonyl modes upon harmonic coupling; (2) the presence of complexes in both *cis* and *trans* geometries [32]; (3) possible variance about the asymmetry of the ring structures; (4) variances of interactions with the solvent. Due to this complexity, we review electronic and IR responses using the same level of theory and describe both structural and dynamic properties of the Cu(L-proline)_2_ complex.

To address the structural properties of the system using optical responses, we consider both the structural flexibility of Cu^2+^(d^9^) coordination in the configurational space [4], and the role of dynamics [6]. Accordingly, first, we adopt quantum ab initio Born–Oppenheimer molecular dynamics (BOMD) [33] to anticipate the structural variances for the reported *cis* and *trans* isomers [32], which present two main structural forms in the configuration space of the system. Using BOMD, we characterize structural fluctuations of prolines, address the role of apical water coordination, and explore aqueous clustering about the polar moieties of the complex. According to the results obtained, we suggest a series of model systems under different degrees of hydration. Next, we use density functional theory (DFT) and time-dependent DFT (TD-DFT) [34,35,36] to test the role of the first neighbor aqueous states searching for convergence with experimentally detected UV-VIS absorption, UV-VIS optical activity, steady-state, and time-resolved mid-IR spectra. 

To understand the validity of the approach, it is important to note, that due to the challenge of describing the relative arrangement of ligands bound to the Cu^2+^(d^9^) cation, the sensitivities of both chiral and IR diagnostics are valuable. However, IR (vibrational) and electronic (chiral) spectral responses describe the relaxation of different polarizations, whose time-range and sampling capacity are different. Obtaining an adequate description of the transition metal cation’s quantum nature according to its coordination is feasible only if we combine the results of both spectral diagnostics. From the obtained structural insights, we demonstrate that the combined experimental and computational approach offers important clues to address the role of Cu^2+^ coordination in metalloenzymes [9,13,14,15], chaperones [10,11,12], and protein misfolding [20,21,22].

## 2. Materials and Methods

The preparation and characterization of the optical absorption and optical rotatory dispersion of the Cu(L-proline)_2_ complex was reported in a previous contribution [24]. We collect steady-state IR spectra of complexes in deuterium oxide using a Nicolet 6700 FTIR spectrometer, Thermo Fisher Scientific Co., Madison, WI, USA. To probe structure and dynamics of the carbonyl moieties of the complex in the electronic ground state, we use a two-dimensional IR (2DIR) femtosecond spectrometer [37], detecting induced IR absorption as a function of time-delay after pump pulses (of about 50 μJ) with the wavelength of the spectral envelope centered at 600 nm, and with a spectral width of about 400 cm^−1^.

To characterize structural variances of the Cu(L-proline)_2_ isomers under different degrees of hydration, we employ ab initio molecular dynamics (MD) simulations as implemented in the CP2K program [33]. Specifically, to account for the physics of the complex, where the Cu^2+^(d^9^) ion has one unpaired electron and the quantum physics of hydrogen bond dynamics, we perform BOMD simulations, up to 0.5 ps long, in the constant-volume constant-energy (microcanonical) ensemble. To characterize a range of structures, we simulate complexes in a cubic box of 17 Å side-length under periodic boundary conditions using generalized gradient approximation Perdew–Burke-Ernzerhof functional, dzvp-molpot basis, and gth-pbe pseudopotentials [33] under the hybrid Gaussian (linear combination of atomic-like orbitals, LCAO) plane wave density functional scheme. The time-step is set to 0.2 fs. Target accuracy for the self-consistent field convergence is 10^−6^. The cut-off and the relative cut-off of the grid level are set to 400 and 100 Rydberg, respectively. The energy convergence threshold is set to 10^−12^. The number of additional molecular orbitals for each spin is set to 100. We compute structural variances for the *cis* and *trans* isomers of the complex [32] under various degrees of hydration, identified as follows: (a) *trans*-0 and *cis*-0 are the complexes without explicit water; (b) *trans*-1 and *cis*-1 are the complexes with one explicit water molecule coordinating apically at the Cu^2+^ ion; (c) *trans*-H and *cis*-H are the complexes where the Cu^2+^ ion and the proline atoms are coordinated with 15 inter-connected first neighbor explicit water molecules; (d) *trans*-W and *cis*-W are the complexes where the Cu^2+^ ion and the proline atoms are coordinated with inter-connected first neighbor and second neighbor water molecules for a total of 29 molecules. In Figure 1, we report a graphical representation of the simulated systems. In the Appendix A, we describe the potential and kinetic energy changes during thermal equilibration before starting structural sampling, as well as the structural properties sampled along the trajectories for these structures. 

According to the results of BOMD simulations (see Appendix A), we choose the *cis*-0, *trans*-0, *cis*-1, *trans*-1b, *cis*-H, and *trans*-H structural cases for spectral studies using DFT and TD-DFT, as implemented in the Gaussian 09 program [38]. The well-hydrated *cis*-W and *trans*-W systems are excluded from spectral studies, as in such large systems the excited state optimization and anharmonic analysis are not yet feasible on a reasonable time using contemporary computational capacity. For these calculations, the copper in the complexes is in a doublet spin state, and hence, we employ the unrestricted B3LYP functional [34]. Considering results reported earlier [35,39], we adopt the LANL2DZ basis set for the Cu^2+^(d^9^) ion, whereas the 6-31++g(d,p) basis set is used for all other atoms. A DFT analysis is performed on structures extracted from BOMD simulations of the complex, either explicitly hydrated or not, optimized using the polarizable continuum model (PCM) [40] to account for bulk solvent effects, which is particularly valuable in structures where first neighbor water molecules are accounted explicitly. The Appendix A provide atomic coordinates for such structures optimized in the ground and the first excited electronic states.

We describe the nature of detected IR absorption in the frequency range from 1200 to 1800 cm^−1^ using a normal mode analysis. The computed frequencies have been scaled by 0.97 to account for the known frequency overestimate featuring DFT calculations. A discussion of structural distributions using nonlinear 2DIR spectral responses relies on the simulation of the 2DIR spectra using the approach reported previously [41]. Specifically, first, we perform a phenomenological fit of the diagonal component of the 2DIR responses using a frequency fluctuation function expressed as $\xi \left(t\right)=\delta \left(t\right)/T_{2}^{*}+\Delta_{1}^{2}\;exp\left(-t/\tau_{c}\right)$. Here, $T_{2}^{*}$ is the pure dephasing time, $\Delta_{1}$ is the frequency fluctuation amplitude of the diffusion term, and $\tau_{c}$ is the correlation time. Other quantities enter the functional expression used in the fit [41]. They are ω_01_ and ω_12_, namely the frequencies of the vibrational transitions |0〉→|1〉, and |1〉→|2〉, respectively. Second, we compute cubic and quartic anharmonic constants for representative complexes in *cis* and *trans* geometries. The anharmonic constants are computed calling the g09 “freq” function with options: “anharmonic”, “selectanharmonicmodes”, and “readanharm”. The anharmonic constants are printed under a tolerance of 10^−8^. Having the anharmonic constants, we evaluate the diagonal and off-diagonal anharmonic frequency shifts following the perturbative approach [42]. Third, we use the fitted frequency fluctuation function, calculated anharmonic frequency shifts, and computed angles between transition dipole moments of the normal modes to plot the line-shapes of the diagonal and off-diagonal resonances of the representative systems, as extracted from the BOMD simulation trajectories, according to the cumulant expansion approach [43].

To address the optical electronic properties of the considered systems, we use time-dependent DFT. Following the approach, we compute amplitudes of optical rotations and model circular dichroism (CD) resonances assuming Lorentzian line-shapes. Furthermore, we compute ORD spectra applying the Kramers–Kronig transform for the modelled CD resonances [44,45]. We attune the spectral width of the Lorentzian line-shapes to match the shape of the experimentally detected ORD spectrum [24].

While TD-DFT provides a description of the electronic excited states, typically, results are unambiguous for transitions between the highest occupied molecular orbital and lowest unoccupied molecular orbital for simple molecules only. For complex species, computed electronic transitions may involve several molecular orbitals without a single dominant component. To reduce the complexity for the complexes considered, we exploit Koopmans’ theorem [46] and search for a transformation of the density matrix [47] to consolidate electronic redistributions specific to a selected transition as the “lower” and the “upper” orbital components. Pairs of such orbital components form the so-called natural transition orbitals (NTO) [48]. By visualizing the NTOs of the electronic transitions responsible for optical density in the visible spectral range, we obtain a systematic and rigorous description of the electronic components governing spectral responses in the visible spectral range in dependence on conformation and degree of hydration.

## 3. Results and Discussion

### 3.1. UV-VIS Optical Activity on Structure and Coordination

In Figure 2a,b, we present optical absorption and ORD spectra for the Cu(L-proline)_2_ complex in the visible spectral range. According to early X-ray [30,31] and electron paramagnetic resonance studies [49], the observed broad optical response in the visible spectral range is assigned to a B_1g_(x^2^ − y^2^)→E_g_(xz,yz) doublet at 590 nm and a B_1g_(x^2^ − y^2^)→B_2g_(xy) transition at 605 nm. At the same time, a magnetically forbidden transition to the A_1g_(z^2^) state has been anticipated at about 1000 nm [50].

To review the experimental optical absorption and ORD spectra in the visible spectral range and to discuss the structural properties of copper proline complexes, we use BOMD simulations and DFT calculations, as described in the Materials and Methods. Here, we start from the energetics of the simulated systems. BOMD trajectories suggest that the *cis*-0 and *cis*-1 systems are slightly more settled than their *trans* analogues, but the energy differences are typically less than 0.01 E_h_: within kT at 300K (see Appendix A). DFT energy minimizations suggest that *trans*-0, and *trans*-1 systems are slightly more energetically favorable than their *cis* analogues, though the energy differences are less than the thermal bandwidth. For more hydrated systems, both BOMD simulations (see Appendix A) and DFT calculations predict energies for the *cis*-H and *cis*-W complexes lower than for the *trans* analogues. However, the energy benefit under the *cis* geometry is only due to more compact hydrogen bonded networks next to the carbonyls of the complex when they are at the same side under the *cis* geometry (see the convex hull presentations in Figure 1). The less complex aqueous distribution next to the *cis* structure suggests a lower configuration entropy for the case. Considering the anticipated differences in hydration, it is possible to expect that such should affect the optical responses, the modeling of which may help in the deduction of the structure. According to the results of the BOMD simulations (see Appendix A), we adopt the relevant *cis*-0, *trans*-0, *cis*-1, *trans*-1b, *cis*-H, and *trans*-H structural cases to conduct theoretical spectral studies as we describe in the Materials and Methods. The case *trans*-1a is not considered as important because theory predicts that in the well-hydrated systems, apical coordination of water with the copper ion happens on the side of CH terminals of the proline rings (as in the *trans*-1b case). 

First, we optimize structures of the six selected structures in the ground and first excited electronic states: see the summary of the geometric properties in Table 1. Next, we calculate the optical absorption and ORD spectra in the visible spectral range: see Figure 2a and Figure 2b, respectively. The computed electronic dispersions demonstrate strong dependencies on the structure. Specifically, we see that the wavelengths of the first three electronic resonances for the *trans*-0 {473 nm, 533 nm, 549 nm}, and for the *cis*-0 conformations {504 nm, 521 nm, 545 nm} are too blue compared to the experimentally observed resonances. Furthermore, the relatively high intensities for the longer wavelength transitions (among the list) disagree with both the experimental results and with the early descriptions [30,31,49,50].

The addition of a coordinating water molecule to the copper ion improves the computed optical absorption and ORD spectra in the visible: electronic transitions shift to the red and the intensities start to resemble the experimental data. However, in the *trans*-1b case, the main electronic transition at 530 nm is still too blue, while the ORD spectra of the hydrated *cis* structures show positive features below 550 nm, which are inconsistent with the experimental results. The TD-DFT calculations predict that only the *trans*-H structural case demonstrates optical absorption and ORD spectra that agree reasonably well with the experimental results. Exploring the ORD spectra computed for the *trans*-H and *cis*-H structures, we observe that the structural distinction is possible due to the sign and intensity of the optical rotation for the second transition in both cases. To proceed further, here, it is important to note that the computed electric and magnetic transition dipole moments, specific to the optical excitation with respect to the structures (see Figure 2), are particularly helpful to contrast the differences in electric and magnetic components in the complex due to the *cis* and *trans* geometry. The relative orientations of the vectors determine the character of the ORD response: the sign of an ORD component is positive if the angle between the two vectors is less than 90°. The sign of the ORD dispersion wing at the higher frequency side is positive if the angle between the two vectors is smaller than 90°. Comparing orientations of the electric and magnetic vectors for the second transitions for the two geometries, we see that the electric component is along the O-Cu-O structural component of the *cis* system. At the same time, for the *trans* geometry, the electric transition moment departs from that direction toward the N-Cu-N structural component. Comparing the NTO and the transition vectors for the second d-d resonance of the *trans*-1 and *trans*-H systems, the departure of the electric displacements toward the N-Cu-N structural component is possible only upon effective admixing of nitrogen electrons with the d-orbital of copper. The theory predicts this only for the *trans* geometry and in the presence of a sufficient number of first (partially second) neighbor water molecules.

Next, we explore the character of the electronic transitions provided by the computed optical absorption and ORD spectra, as shown in Figure 2. Specifically, in Figure 3 we report NTO pairs, which contribute to the three lowest energy electronic transitions for the selected structural cases. In all cases, excitation in the visible spectral range brings molecules into the same NTO, where the $d_{x^{2}-y^{2}}$ component of Cu^2+^ is in an antibonding arrangement with the π* antibonding component of the ligands. Because of this, in Figure 3, we image three NTO pairs, which share the same upper state, while their lower (departure) orbital components are along the descending energy ladder, according to the frequencies of resonances of the computed optical transitions.

Our TD-DFT calculations predict that d-d transitions between $d_{x^{2}-y^{2}}$, $d_{z^{2}}$, $d_{xz}$, and $d_{yz}$ states dominate the optical properties of the complex in the visible spectral range. This is consistent with the results of previous publications [31,49,50], save for the discussion of EPR spectra in Cu(proline)_2_ single crystals that suggested $d_{x^{2}-y^{2}}$ orbital to be the ground state [49]. Our experimental and theoretical studies for the complex in solution predict the energy ladder, which is typical for a Cu^2+^(d^9^) ion in an octahedral electric field under the Jahn–Teller distortion [51]. The agreement is clear for the *trans*-H structural case, where a relatively good hydration is accounted for explicitly. In this case, the excitation by the first pair accounts for the departure from the orbitals where the $d_{z^{2}}$ component of Cu^2+^ is in antibonding arrangement with the π component of oxygen from water. Furthermore, the departure orbitals for the second and third transitions account for the $d_{xz}$ and the $d_{yz}$ contributions of the Cu^2+^ ion, respectively. Apparently, the predicted small intensity of the lowest energy transition (for example, the absorption at 790 nm for *trans*-H in Figure 2a) is due to the symmetry relaxation of the complex rings and slight admixing of ligand contributions in the first NTO pair lower states of *trans*-H and *cis*-H structures (see Figure 3).

Here, we note that the NTO description helps to monitor the electronic effects associated with the considered electronic transitions. For example, we see that the introduction of explicit water apically coordinated to Cu^2+^ is essential to adequately model the field splitting for the lower energy range optical electronic transitions: for example, see the relative positions of the lower state of the second NTO pair for the *trans*-0 and of the first NTO pair lower states for the *trans*-1a and *trans*-1b. In addition, this affects the character of the spatial distributions of the electronic components: for example, see the difference in the shapes of the first NTO lower state’s $d_{z^{2}}$ components of the *cis*-0 and of the *cis*-1 structures. Further, we note that the introduction of water affects geometry as well (see Table 1). When explicit water is added, bonds in the complexes are longer, and the rings form a less planar structure. The latter reduction would relax symmetry rules for IR absorption. Accounting these facts is important for further efforts in modelling.

### 3.2. IR Diagnostics on Intra- and Inter-Molecular Correlations

Having performed a structural analysis using electronic optical responses, it is important to review the IR properties of the system, and mechanisms of the inhomogeneity of the IR absorption of carbonyl modes of the Cu(L-proline)_2_, in particular [24]. In Figure 2c, we compare the IR absorption of the modelled structures computed by DFT calculations with FTIR spectrum for the prepared Cu(L-proline)_2_ complex. First, it is worth noting that the calculated spectra show a significant sensitivity of the normal modes on the degree of hydration. The frequency separation between the COO/C-H scissor vibrations (1400 cm^−1^) and carbonyl modes (1600 cm^−1^) is smaller for more hydrated systems. Second, the results of calculations show that the frequency split between the symmetric and anti-symmetric carbonyl modes upon harmonic coupling cannot explain the overall width of the band at 1600 cm^−1^. At the same time, the IR responses in the considered spectral range do not help to distinguish whether we deal with the *trans* or *cis* geometries of the complex. Overall, the computed IR responses of the *trans*-H and *cis*-H structures both demonstrate reasonable agreement with the experimental FTIR.

To review the anticipated structural distribution, we employ time-resolved nonlinear 2DIR spectroscopy, following procedures reported earlier [37]. In particular, searching for the optimal time-delay to sample 2DIR, first, we use broad-band IR-pump/IR-probe to measure the population relaxation of carbonyl vibrations in the complex under the magic angle, which helps to remove ambiguities due to orientation dynamics. The signal under the magic angle decays with two lifetimes: *τ*_1_ = 380 fs (a_1_ = 0.5) and *τ*_2_ = 820 fs (a_2_ = 0.5). For a comparison, population relaxation of the carbonyl of a free proline in water (data not shown), detected under the same conditions, can be fitted with two similar components: *τ*_1_ = 340 fs (a_1_ = 0.8) and *τ*_2_ = 1100 fs (a_2_ = 0.2). The anisotropic decay of the carbonyl response of the complex follows a single exponential decay: *τ*_R_ = 5 ± 1 ps. This is two times slower than for the proline in deuterium oxide, where we determine *τ*_R_ = 1.75 ± 0.25 ps. The broad-band time-dependent IR does not show any spectral dynamics. Instructed by the ultrafast broad-band experiment, next, we measure the 2DIR spectra of the carbonyl moieties of the complex in deuterium oxide at 0.9 and 2 ps; see Figure 4b,c. The 2DIR response has a quasistatic character on the time-scale of 2 ps: there are no obvious changes in shape. Additionally, one may see that the spectra contain noticeable cross-peaks. Since we do not observe significant differences under the perpendicular polarization setting, we focus on the analysis of 2DIR spectra detected under a parallel polarization setting, only.

To characterize the dynamics, we make a fit of the detected 2D line-shapes using a phenomenological frequency fluctuation correlation function [41]. Here, we notice that, regardless of the geometry or the degree of hydration, any structural case is expected to demonstrate both a lower frequency intense carbonyl antisymmetric stretching and a higher frequency weaker symmetric mode (see Figure 2c). Therefore, if we assume that the two structural cases (namely, *cis* and *trans* conformers) provide the experimentally observed inhomogeneous width of the band at 1600 cm^−1^ (one to express at the higher and another to dominate at the lower frequency side), we would have to fit four resonances, because each structure should have their own pair of symmetric and antisymmetric stretching. However, if such is the case, within the considered bandwidth, the weak antisymmetric mode of the low-frequency system would overlap with the strong symmetric mode of the high-frequency system. Owing to the overlap, the fit of the observed 2D line-shapes cannot be completed rigorously using four resonances. To deal with the fitting practically, we use three diagonal resonances, where the line-shape of the intermediate frequency resonance would account contributions of the two spectrally overlapping modes (the weak antisymmetric mode of the low-frequency system and the dominant symmetric mode of the high-frequency system). Figure 4d–f and Table 2 present the fitted properties of the three diagonal resonances. The dominant spectral signature at ω_01_ = 1604 cm^−1^ (fitted as one resonance) is interpreted to be the overlap, as we discuss. The fitted properties determine the frequency fluctuation correlation function, which we can use to express the 2DIR spectra of stretching modes specific to the model systems that we have computed using DFT. Additionally, in Figure 4h,i, we present the error 2D spectra (difference between the fitted and the experimental 2DIR line-shapes), which indicate very weak cross-peaks in the off-diagonal regions. This is possible since we subtract from the experimental data the 2DIR spectra computed using diagonal resonances only. Evidence of the low amplitude cross-peaks in the error-differences suggest very weak but present anharmonic couplings between the antisymmetric and symmetric modes specific to the structures explored.

Next, using the frequency fluctuation function resulting from the fit [41], we model 2D line-shapes for the representative structural cases (as shown at the right side of Figure 2) according to the DFT results: see Materials and Methods. Figure 5a,f,b,g compare the 2DIR spectra of the *cis*-H(*trans*-H) and *cis*-1(*trans*-1) forms, respectively. Because the experimental 2DIR (Figure 5d) and the difference spectrum computed at 0.9 ps time-delay (Figure 4h) do not show intense cross-peaks, there is a bias against such model systems, where DFT studies suggest larger anharmonic interactions between carbonyl vibrations. Specifically, the less hydrated case *cis*-1 is predicted to demonstrate intense cross-peaks between its antisymmetric and symmetric carbonyl modes (red arrow in Figure 5b). To model the full width of the 2DIR spectrum for the considered *cis* structure, we mix the 2DIR contributions of the *cis*-H and *cis*-1 cases in the ratio 3:1, as shown in Figure 5c, to match the full diagonal width of the 2DIR as in the experiment (Figure 5d). Even under a smaller weighting, the intense cross-peak of *cis*-1 (see red arrow in Figure 5c) is inconsistent with the experimental results. Comparatively, in Figure 5f,g,e we present 2DIR and linear IR signals calculated for the *trans*-H and *trans*-1 model systems, and the sum of their contributions (under the ratio 3:1, respectively) demonstrate a 2DIR line-shape comparable to the experiment. 

Let us review the results of the above discussion concerning the optical electronic and IR responses. In particular, the results of our study of the spectral responses in the visible spectral range (Figure 2a,b) suggest that the *trans* geometry of the complex in aqueous solution is dominant. The analysis of the IR spectral responses (Figure 5) calls for a not irrelevant heterogeneity of the degree of hydration. As a result, we suggest that the red edge and the central part of the IR response of carbonyl moieties are due to the dominant contribution of the antisymmetric and symmetric stretching modes of relatively well-hydrated *trans*-H-like structures. The carbonyl modes of less optimally hydrated structures (analogous to the case *trans*-1) would contribute to the inhomogeneous broadening at the higher frequency side of the IR band centered at about 1600 cm^−1^. It is remarkable that the optical electronic and IR responses are both inconsistent with a detectable presence of *cis* conformers in the solution. In this description, we evaluate the relative contributions of the two conformations, giving less importance to the weights of the three phenomenologically fitted resonances: see Table 2 and Figure 4. The choice agrees with the conclusions of the analysis of the ORD spectrum in Figure 2b and the outcome of the higher-level theory 2DIR analysis as presented in Figure 5, as the main purpose of the fit presented in Figure 4 is to anticipate the character of the frequency fluctuation correlation function.

### 3.3. Photo-Dynamic Perspective of Copper Electronics and Coordination

Having completed a structural analysis while sampling electron optical and IR response functions, we explore structural and electronic properties of the complex in water in the presence of radiation resonant with the d-d transitions, as we describe in Figure 2 and Figure 3. To do this, we use visible pump/IR probe spectroscopy to detect changes in vibrational dynamics upon excitation of the complex into the electronic excited state. Specifically, in Figure 6a, we present mid-IR spectral responses induced by visible pulsed excitation at 600 nm and detected at 0.5 and 1.5 ps time-delays. Here, we disregard responses at the early time-delay (within the first 0.5 ps), as signals are complicated with a coherent artefact [52]. The induced IR response shows a positive spectral signature at 1570 cm^−1^, and two bleach components in the higher frequency spectral region, between 1585 and 1640 cm^−1^. We may assign the positive signature to the TD-DFT predicted antisymmetric stretching mode of the main *trans*-H structure when in the first excited electronic state: see the red line spectrum in Figure 6b, or the red line spectrum for the *trans*-H structure in Figure 2c. We now consider the spectral bleach contributions. The spectral width of the holes is about 16 cm^−1^. This is slightly narrower than the homogeneous width (20 cm^−1^) in the 2DIR spectrum, as shown in Figure 4. What is interesting is that the lower frequency bleach at 1590 cm^−1^ is nearly as intense as the higher frequency one at 1620 cm^−1^. In light of the already developed structural description, the frequency difference and the comparable intensities lend additional support to the hypothesis of a heterogeneity involving hydrogen bonding. Furthermore, since the two bleaches are relatively well-separated, we may consider a non-smooth variance of the degree of hydration of the carbonyl moieties. Further, the results of the BOMD simulations (see Appendix A), indicate that, due to its structure, the water molecule, which apically coordinates the Cu^2+^(d^9^) ion, cannot participate identically with the aqueous clusters next to the two carbonyls. This is particularly obvious in the arrangement of the complex in the *trans* geometry.

### 3.4. Structural Flexibility and Implications

Accomplishing the structural analysis, we are left with the initial inquiry of if the selected complex may help to understand structural flexibility specific to Cu^2+^(d^9^) metalloproteins [9,13,14,15], and systems where chaperone folding or misfolding play central roles [20,21,22]. Here, we review the structural tendencies of Cu^2+^(d^9^) in the Cu(L-proline)_2_ complex accounting for modulations as anticipated upon excitation of d-d transitions. In this respect, it is instructive to return to the structural properties of the *trans*-H system in the ground electronic state and in the first excited electronic state.

For the *trans*-H structural case, DFT predicts that on electronic photoexcitation, the angle NO′CuN′ should decrease from 157° to 142°, and the length of the bonds between Cu^2+^ and the proline atoms increase by about 0.15 Å (see Table 1). These structural changes correlate with the Mulliken charge and spin density changes on the metal ion from −0.82 and 0.57 to −1.86 and 0.6, respectively. Thus, the theory suggests that on photoexcitation there is partial reduction of the metal ion that correlates with the structural tendency of the complex to shift toward tetrahedral geometry. The analogous data for the *cis*-H system suggest more conservative behavior. Considering the experimental and theoretical data, we should not expect strong reactive and restructuring tendencies for the metal ion in the systems, but effective restructuring of the aqueous networks which involve the axial water ligand. In fact, after excitation with a pulsed radiation at 600 nm, there is a bleach signal at the frequency of the OD stretching mode to gain its maximal value at 0.8 ps and to decay before 1.4 ps: (see Appendix A). Since there are no spectral dynamics in the spectral region of the OD stretching modes, we ascribe the bleach to the weakening and recovering of hydrogen bonds in the aqueous network adjacent to Cu^2+^(d^9^) ions.

Here, it is important to recollect the role of the axial ligand in copper enzyme reactivities [9,13,14,15], and the difference in coordination behavior in chaperones [10,11,12] and in structurally unstable systems [20,21,22]. The Cu(proline)_2_ complex is a simple example that can shed light on mechanisms of bio-copper structural ambivalence and, hence, variations in reactivity. Specifically, structural flexibility may be specific to electronic fluctuations of the copper ion (with its nearly complete electronic shell), which is sensitive to the nature of the axial ligand. While Raman microscopy suggested that copper and zinc are both associated with the histidines of amyloid depositions [53], the two ions demonstrate different characters in competition for prion binding [54]. In contrast to Cu^2+^(d^9^), Zn^2+^(d^10^) with its filled d-shell tends to follow conservative tetrahedral covalency [55,56,57]. The results of our studies suggest the structural and electronic properties of Cu(proline)_2_ to be in critical dependence on modeling the first neighbor interactions explicitly. Considering the role of the axial ligand to exclude or let in water, mobile species, or reactants [23], and the structural flexibility in loose associations of Cu^2+^(d^9^) with amyloid and prion structures [58,59,60], here, we propose that the structural control in copper-containing species is governed, at least in part, by fixing or altering the nature of the axial ligand. This correlates well with the recent reports that either complexing Cu^2+^ with ethylenediaminetetraacetic acid [61] or with 5,7-dichloro-2-[(dimethylamino)methyl]-8-hydroxyquinoline [62] helps to hinder the development of Alzheimer’s disease. It is interesting to add here that absence of an aqueous ligand was reported to be important for divergence from the common proline N,O-chelation pattern in a copper chain polymer [63].

## 4. Conclusions

The structural properties of Cu^2+^(d^9^) complexes with amino acids and polypeptides are interesting due to their presence in naturally occurring biological molecules and promising roles in pharmacological applications. Cu complexes with amino acids may be considered as the simplest models in this group of molecules and structures. In this study, we explore the electronic, structural, and dynamic properties of Cu(L-proline)_2_ under low and high degrees of hydration. An analysis of experimental and computational data shows that explicit consideration of first neighbor water clustering is critically important for the correct description of the electronic properties of the complex. Further, we can deduce that a well-hydrated *trans* conformer is the main structural form of the complex in water, where the antisymmetric stretching mode of the carbonyl moieties in the fully hydrated *trans* conformer dominates the experimentally observed broad IR absorption at 1605 cm^−1^. The inhomogeneous broadening of this IR transition at the blue side may be ascribed to *trans* conformers interacting with the solvent more weakly. The extracted structural tendencies in correlation with electronic properties suggest that structural flexibility/ plasticity specific to Cu^2+^(d^9^) can be used to control local architecture and reactivity in chaperone systems and metalloenzymes.

## Figures and Tables

**Figure 1 molecules-27-05846-f001:**
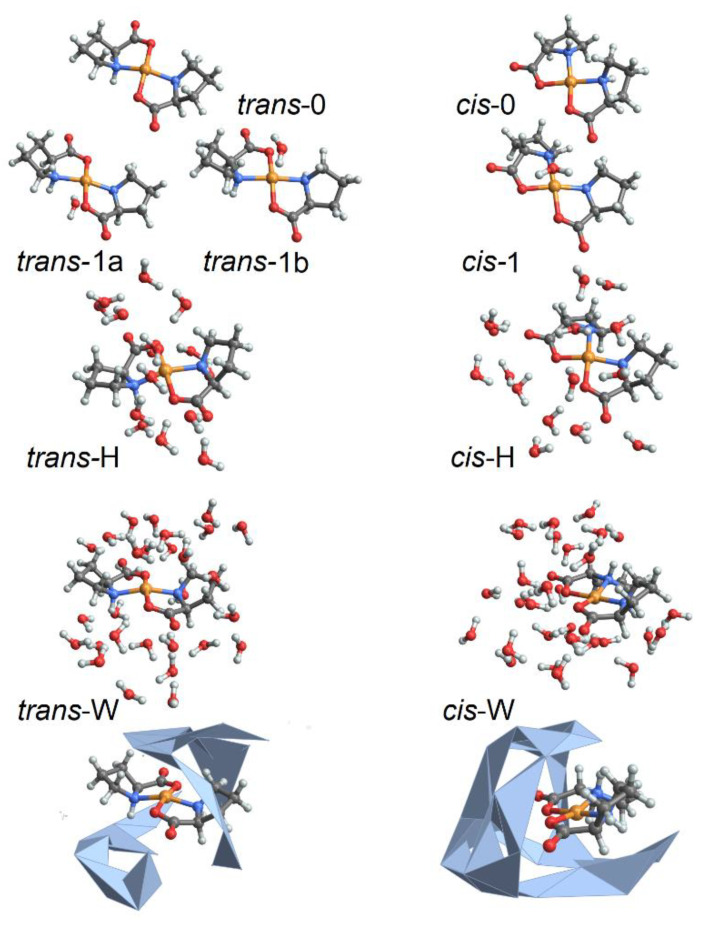
Selected model systems for BOMD studies: without water, structures *cis*-0 and *trans*-0; with 15 water molecules: structural cases *cis*-H and *trans*-H; and with 29 water molecules: structural cases *cis*-W and *trans*-W. Here, to stress the difference in character of water packing next to the complex under the two geometries, we image water in *cis*-W and *trans*-W structures combining wireframe and convex hull formats.

**Figure 2 molecules-27-05846-f002:**
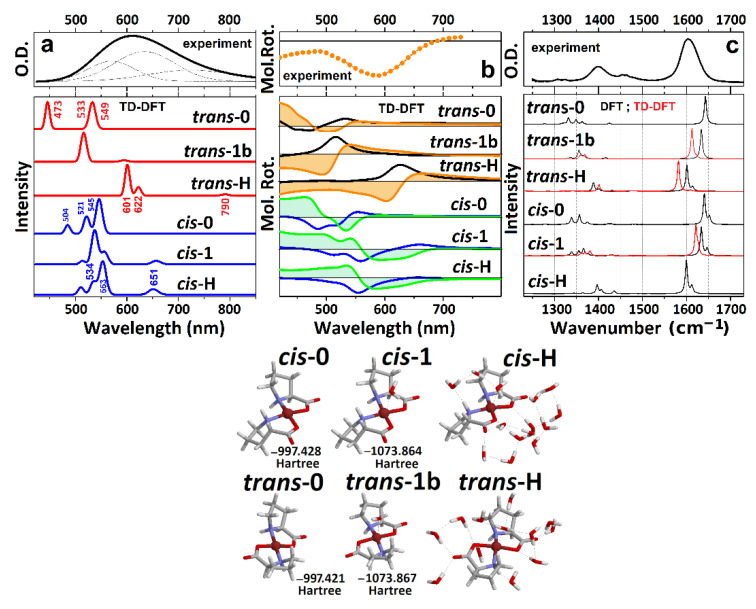
(**a**) Experimental optical absorption spectrum of the Cu(proline)_2_ complex and calculated responses of the model systems, as shown at the right side. For clarity, the spectral dispersions are calculated using convolution with a narrow line shape. (**b**) Experimental ORD spectrum and theoretical CD (black lines correspond to *trans*-complexes and blue lines to *cis*-complexes) and ORD (orange filled profiles correspond to *trans*-complexes and green filled profiles to *cis*-complexes) responses for the model systems calculated as described in Materials and Methods. (**c**) Experimental FTIR spectrum and calculated IR responses for the model systems in the electronic ground state (black lines) and in the first excited electronic state (red lines). For clarity, the spectral dispersions are calculated using convolution with a narrow line shape. Right side: images of the model molecular systems under *cis* and *trans* geometries: numbers next to the structures refer to energies after optimization in the ground electronic state.

**Figure 3 molecules-27-05846-f003:**
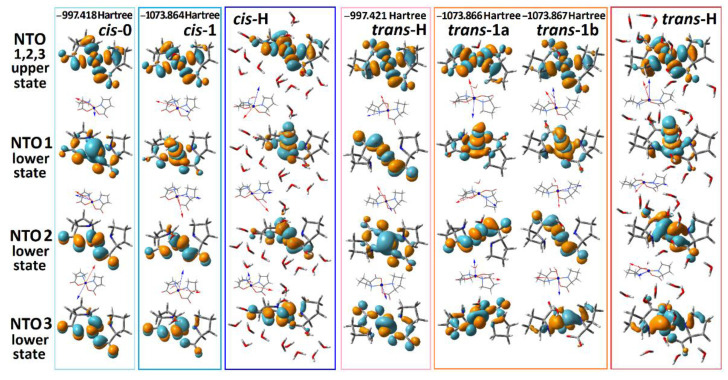
NTO presentation for the three lowest energy optical transitions that provide the optical transitions in the visible for the considered structural cases. Orientations of the computed transition electric (red) and magnetic (blue) dipole moments specific to the optical excitation with respect to the structures: we place the vector descriptions next to the images of the lower (departure) states of the NTO pairs because the three transitions share the same upper (arrival) state.

**Figure 4 molecules-27-05846-f004:**
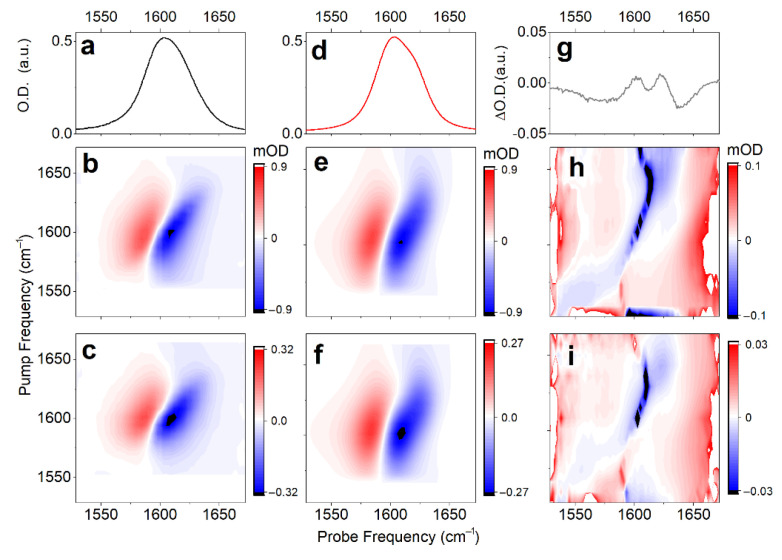
Linear IR (**a**), and 2DIR responses of Cu(L-proline)_2_ complex detected at 0.9 ps (**b**), and 2 ps (**c**) time-delays under parallel polarization. Linear IR (**d**) and 2DIR responses (**e**,**f**) fitted according to a protocol reported previously [41]. (**g**–**i**) Differences between experimental and fitted spectra for the linear IR and 2DIR responses at 0.9 and 2 ps time-delays, respectively.

**Figure 5 molecules-27-05846-f005:**
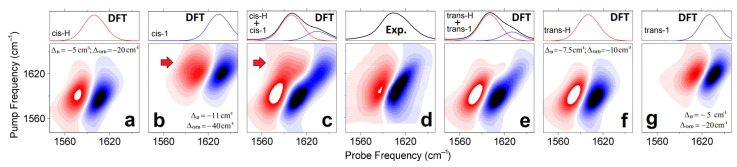
2DIR and linear spectra computed for *cis*-H (**a**), *cis*-1 (**b**), *trans*-H (**f**), *trans*-1 (**g**) model systems. The line-shapes are according to the frequency fluctuation correlation function, the diagonal anharmonic shifts, and the off-diagonal anharmonic shifts calculated as described in the Materials and Methods. (**c**,**e**) sums of the 2DIR contributions computed for the *cis*-H and *cis*-1 cases and for the *trans*-H and *trans*-1 cases in the ratio 3:1. (**d**) 2DIR experimental spectrum of a Cu(L-proline)_2_ complex at 0.9 ps time-delay and FITR spectrum. and calculated 2DIR responses for the selected model systems. Red arrows in panels (**b**,**c**) indicate cross-peaks due to off-diagonal anharmonicity as anticipated for carbonyls in the *cis*-1 structure. Computed linear spectra (red and blue lines) are slightly asymmetric: each line-shape includes both a dominant-intense antisymmetric stretching resonance, and a weak symmetric stretching resonance contributing into a slight broadening at the higher frequency side.

**Figure 6 molecules-27-05846-f006:**
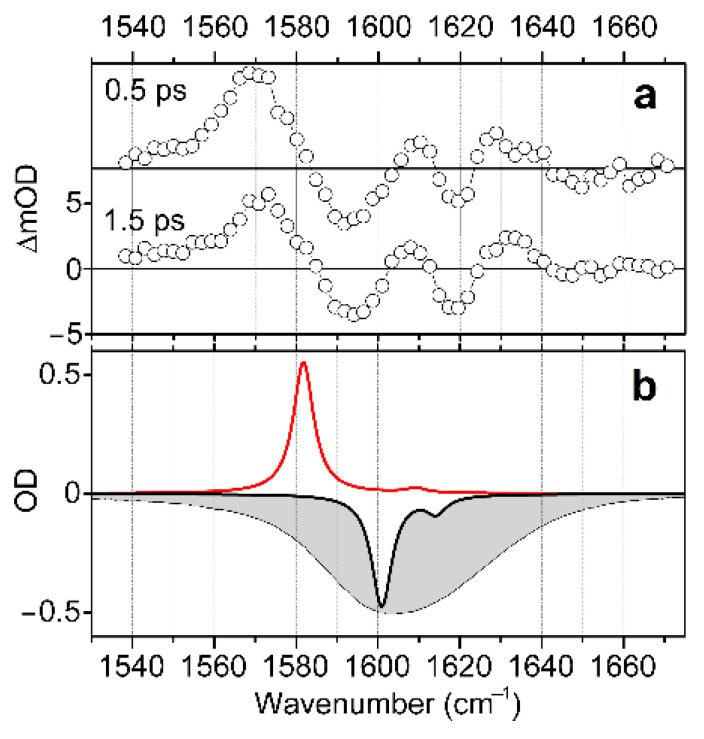
(**a**) Induced IR spectra detected at 0.5 and 1.5 ps time-delays after visible pulsed excitation of d-d transitions at 600 nm. (**b**) Negative FTIR spectrum (grey filled profile), negative calculated IR response of the *trans*-H structure optimized in the ground electronic state (black line), and calculated IR response of *trans*-H structure optimized in the first excited electronic state (red line). The calculated IR responses are expressed using convolution with a narrow line-shape for the purpose of clarity, as in **Figure 2c**.

**Table 1 molecules-27-05846-t001:** Calculated structural properties of the Cu(L-proline)_2_ complexes in the ground electronic state (see Figure 2 for graphical representations of the complexes). In brackets we report the values obtained from the optimization of the first excited electronic state. The presence/absence of the prime on the atomic symbols denotes atoms belonging to different prolines. In the definition of the dihedral angle, A and B represent the Cu^2+^ coordinating O′ and N′ atoms of the *trans* geometry systems, and, correspondingly, the Cu coordinating N′ and O′ atoms of the *cis* geometry systems. O_w_ represents the oxygen atom of the coordinating water molecule.

	*trans*-0	*trans*-1b	*trans*-H	*cis*-0	*cis*-1	*cis*-H
Cu-O/Å	1.95 (2.00)	1.97 (2.08)	2.02 (2.2)	1.96 (2.04)	1.97 (2.02)	2.00 (2.07)
Cu-N/Å	2.03 (2.3)	2.05 (2.19)	2.05 (2.15)	2.1 (2.2)	2.06 (2.25)	2.06 (2.21)
C=O/Å	1.238 (1.241)	1.24 (1.25)	1.25 (1.26)	1.239 (1.242)	1.241 (1.243)	1.25 (1.26)
NCuO′	95° (103°)	95° (97°)	95° (96°)	99° (102°)	98.9° (100°)	98° (100°)
N′CuO′	85° (80°)	85° (81°)	83° (78°)	84° (79°)	84° (81°)	83.5° (80°)
NA′CuB′	178° (168°)	174° (162°)	157° (142°)	178° (171°)	175° (179°)	179° (178°)
Cu-O_w_/Å		2.44 (2.0)	2.33 (1.96)		2.47 (2.02)	2.56 (2.03)
NCuO_w_		97°, 88° (102°, 96°)	113°, 98° (113°, 106°)		102°, 91° (104°, 99°)	103°, 100° (107°, 100°)

**Table 2 molecules-27-05846-t002:** Results of the fit of linear IR and 2DIR experimental spectra using phenomenological frequency fluctuation correlation function $\xi \left(t\right)=\delta \left(t\right)/T_{2}^{*}+\Delta_{1}^{2}\;exp\left(-t/\tau_{c}\right)$ for the case of three vibrational resonances [41]. Here, ω_01_ and ω_12_ are the frequencies of the vibrational transitions |0〉→|1〉, and |1〉→|2〉, respectively; $T_{2}^{*}$ is the pure dephasing time; $\Delta_{1}$ is the frequency fluctuation amplitude of the diffusion term; $\tau_{c}$ is the correlation time. Numerical values in brackets provide the numerical error as described in Ref. [41].

ω_01_/cm^−1^	ω_12_/cm^−1^	Weight	Δ_1_/cm^−1^	*T*_2_*/ps	$\tau_{c}$/ps	γ_1_/ps^−1^	γ_2_/ps^−1^
1595.1(0.1)	1588.5(0.2)	0.2	8.4(0.1)	0.80(0.02)	1.57(0.06)	1(0.01)	1.3(0.01)
1604.1(0.1)	1598.5(0.1)	0.56	8.4(0.1)	0.96(0.01)	1.57(0.06)	0.98(0.01)	1.3(0.01)
1620.9(0.1)	1612.3(0.1)	0.24	8.4(0.1)	0.96(0.01)	1.57(0.06)	1(0.01)	1.3(0.01)

## Data Availability

The data presented in this study is available from the authors on reasonable request.

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
