# Peer review of "Cu(Proline)2 Complex: A Model of Bio-Copper Structural Ambivalence"

_molecules, 2022, doi:10.3390/molecules27185846_

Round 1

Reviewer 1 Report

This manuscript describes a BOMD study on metalorganic copper complexes in combination with spectra simulations. The main objective is to reveal the local coordination of the copper and its hydration in the studied complexes. This is a difficult task as the large amount of literature on this topic demonstrates. The presented work is solid and in principle suitable for publication.

However, a major concern represents the structure selection from the BOMD for the spectra simulations. Because structure assignments (well hydrated trans H-structure) are made from the simulations the question arises how sensitive are the conclusions to this more or less arbitrary selection? Why are the spectra not calculated as ensemble averages, even at the GGA level? This can give some insight into the bias due to structure selection.

Minor points:

·        Is the wave function a LCAO or plane wave expansion?

·        What is the meaning of energy corrections?

·        How long are the BOMD trajectories?

·        Which thermostat was used and how well is the system energy conserved?

·        Why is B3LYP used in combination with the 6-31++g basis when it was optimized for the 6-31G** basis set?

·        How is the dielectric constant for explicit and implicit water adjusted to each other?

·        How is the statement “Such an arrangement for the cis structure would have a lower entropy.” obtained from the presented calculations?

·        Meaning of “Architecture” in the 3.1 subtitle?

Altogether, the manuscript is suitable for publication if the above concerns are appropriately addressed.

Reviewer 2 Report

It is a well written paper in which the author carefully analysed the spectra(ftir, IVVIS) of cu-LPr complexes with or without water. All of the conclusion is clearly proven.

Question:

What was the reason to apply some type of anharmonic correction?

What was the reason to perform BOMD simulation, if the spectra is calculated from 1 configuration? (If it is not true than it would be nice to emphasize better).

I know that the B3LYP is not the worst functional for calculatin UV/VIS spectra, Can you be to some simple calculation with m06 or CAM-B3LYP functional with at least triple zeta quality basis set.

Can you be so kind to comment an possible polymeric formation of CuLP2 as describe in ACT. CRYST 2015 C71 271   

Round 2

Reviewer 1 Report

The authors have improved the presentation of their manuscript! However, they have answered some of my questions only in a general form unspecific to their work presented in the manuscript. Therefore, I cannot recommend the publication of the article without further corrections. In particular, the authors answer to my question about the sensitivity of their results to the more or less arbitrary structure selection form the BOMD simulation is not sufficient to judge the quality of their work. I have no doubts on the validity of (BO)MD simulations and I believe in the ergodicity hypothesis. However, the specific question for this work is how sensitive are the results to the structure selection? The authors should answer this within the used formalism if the structure extraction agrees with the simulated configuration space.

In this respect, I am also wondering how representative a configuration space of a 0.5 ps! trajectory can be. Thus, the authors might reconsider the title of their work. Is this study really representing the “dynamics” of a complex?

Minor points:

·      It seems my basis set question was not well formulated. The question is: Why are you using the 6-31G++ basis set in combination with the B3LYP functional, whereas this functional was optimized for the 6-31G** basis set. Is this related to Rydberg excitations?

·   Is the use of explicit water in PCM really recommended in the literature?
